# TAGET: a toolkit for analyzing full-length transcripts from long-read sequencing

Yuchao Xia[1,2,10], Zijie Jin [3,4,10], Chengsheng Zhang[2,10], Linkun Ouyang [5,10], Yuhao Dong[2], Juan Li [6], Lvze Guo[2], Biyang Jing[2], Yang Shi [7], Susheng Miao [8] ✉ & Ruibin Xi [4,5,9] ✉

Single-molecule Real-time Isoform Sequencing (Iso-seq) of transcriptomes by PacBio can generate very long and accurate reads, thus providing an ideal platform for full-length transcriptome analysis. We present an integrated computational toolkit named TAGET for Iso-seq full-length transcript data analyses, including transcript alignment, annotation, gene fusion detection, and quantification analyses such as differential expression gene analysis and differential isoform usage analysis. We evaluate the performance of TAGET using a public Iso-seq dataset and newly sequenced Iso-seq datasets from tumor patients. TAGET gives significantly more precise novel splice site prediction and enables more accurate novel isoform and gene fusion discoveries, as validated by experimental validations and comparisons with RNA-seq data. We identify and experimentally validate a differential isoform usage gene *ECM1*, and further show that its isoform ECM1b may be a tumor-suppressor in laryngocarcinoma. Our results demonstrate that TAGET provides a valuable computational toolkit and can be applied to many full-length transcriptome studies.

The development of RNA sequencing based on second-generation sequencing (RNA-seq) technologies has revolutionized transcriptomics studies and has been widely used for various transcriptome analyses such as gene or isoform expression quantification, alternative splicing analysis, and gene fusion detection[1–3]. However, the read length of RNA-seq is short relative to the transcript length, and the majority of short reads can only cover a small portion of transcripts. This short read length causes ambiguity in alignment of short reads to isoforms and makes full-length transcript analyses difficult. Although advanced computational tools have been developed to address this problem[4–6], it is often more desirable to have longer reads for more comprehensive analysis of full-length transcripts. The RNA sequencing platform Iso-seq of PacBio, based on the HiFi sequencing,

produces long and accurate reads[7]. Iso-seq has an N50 read length ~2500 bp and achieves an accuracy of 99.9%. Considering that the average transcript length is around 1300 bp, Iso-seq provides an ideal platform for full-length transcriptome analysis.

A number of computational tools have been developed for long-read sequencing data[8–12]. However, integrated computational frameworks for analyzing Iso-seq data are still lacking. In addition, current alignment tools are not optimized for Iso-seq data and the alignment quality needs to be further improved. It is well-known that over 99% of the splice junctions are canonical junctions with highly conserved dinucleotides for the donor and receptor sites, such as GT and AG, respectively[13]. We observe that over 40% of the predicted novel junctions by the current alignment tools are non-canonical, indicating that

[1]College of Science, Beijing Information Science and Technology University, 100192 Beijing, China. [2]Beijing GeneX Health Co.,Ltd, 100195 Beijing, China. [3]Peking University International Cancer Institute, Health Science Center, Peking University, 100191 Beijing, China. [4]School of Mathematical Sciences, Peking University, 100871 Beijing, China. [5]Academy for Advanced Interdisciplinary Studies, Peking University, 100871 Beijing, China. [6]Department of Biomedical Engineering, College of Future Technology, Peking University, 100871 Beijing, China. [7]BeiGene (Beijing) Co., Ltd., Beijing, China. [8]Department of Head and Neck Surgery, Harbin Medical University Cancer Hospital, 150081 Harbin, China. [9]Center for Statistical Science, Peking University, 100871 Beijing, China. [10]These authors contributed equally: Yuchao Xia, Zijie Jin, Chengsheng Zhang, Linkun Ouyang. ✉e-mail: drmiaosusheng@126.com; ruibinxi@math.pku.edu.cn

many novel junctions are false positives. Further, transcriptome quantification analyses rely on accurate transcript annotation and classification. Different isoforms can have shared junctions, which can cause ambiguity for isoform quantification and requires accurate transcript annotation and classification to avoid the resulting false discoveries in downstream quantification analyses.

In this paper, we present a Toolkit for Analyzing full-length GEne Transcripts (TAGET) for Iso-seq. This tool consists of four components, including transcript alignment, annotation, quantification, and gene fusion detection. Quantification includes gene/isoform expression quantification, differential expression gene (DEG) analysis, and differential isoform usage (DIU) analysis. We assess the performance of TAGET using a public Iso-seq[14] as well as seven pairs of newly sequenced Iso-seq data from matched cancer and normal tissues. We find that TAGET demonstrates superior performances on transcript alignment, annotation, expression quantification, and gene fusion detection. Experimental validation of novel transcripts and gene fusions shows that around 86% of novel transcript candidates and gene fusion candidates can be validated. Further, using TAGET, we identify an important DIU gene *ECM1* and experimentally validate the tumor suppressor role of one isoform of *ECM1*. Finally, we apply TAGET to ONT data and find that TAGET also performs well.

## Results

### TAGET is an integrative toolkit for transcript alignment, annotation, and expression quantification

TAGET uses polished high-quality transcripts in fasta format as input (Fig. 1) for full-length transcriptome analysis. Following the Iso-seq data analysis protocol, TAGET only considers transcripts supported by at least two circular consensus sequences (CCS). TAGET first aligns transcripts to the reference genome by integrating alignment results from long and short reads and further improves splice site prediction using Convolutional Neural Network (CNN). TAGET then annotates transcripts by comparing with reference transcript databases and classifies them into seven classes as defined previously[10]. Finally, TAGET estimates the expression of genes and isoforms and performs downstream analyses such as DEG and DIU analysis.

Mapping methods can be classified as long-read mapping and short-read mapping methods. The long-read mapping algorithms such as GMAP[15] and minimap2[9] can map full-length transcripts to the reference genome. However, we find that these long-read mapping algorithms tend to maximize mapping continuity when trying to maintain the mapping accuracy. This strategy can lead to the merging of two separate exons if one of the exons is considerably shorter than the other (Supplementary Fig. 1a). On the other hand, short-read

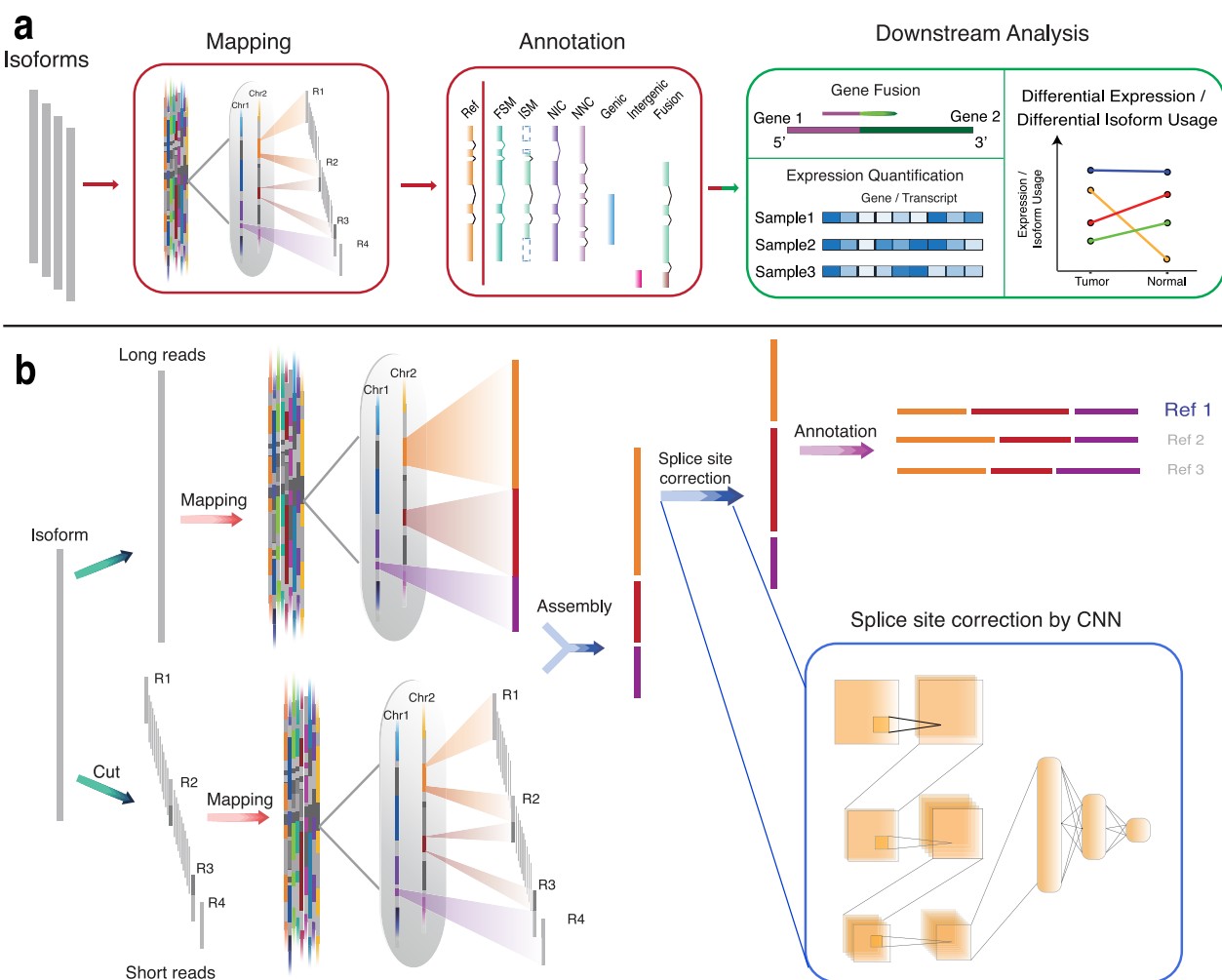

**Fig. 1 | TAGET workflow. a** TAGET first maps the long-reads to the reference genome by assembling long-read and short-read mappings. Then, TAGET annotates the transcripts by comparing with reference isoform databases. The transcripts are classified into seven types (FSM, ISM, NIC, NNC, Genic, Intergenic, and Fusion). TAGET finally performs downstream analyses based on the mapping and annotation. **b** The mapping steps of TAGET. Given a long read, TAGET first cuts it to overlapping short-reads. The long-read and obtained short-reads are mapped to the reference genome using long-read and short-read mapping algorithms, respectively. For the subsequences that have inconsistent long-read and short-read mappings, TAGET collects the mappings of each nucleotide in this subsequence and determines the best assembly of the mappings of the nucleotide. Finally, CNN is applied to locally adjust the splice sites.

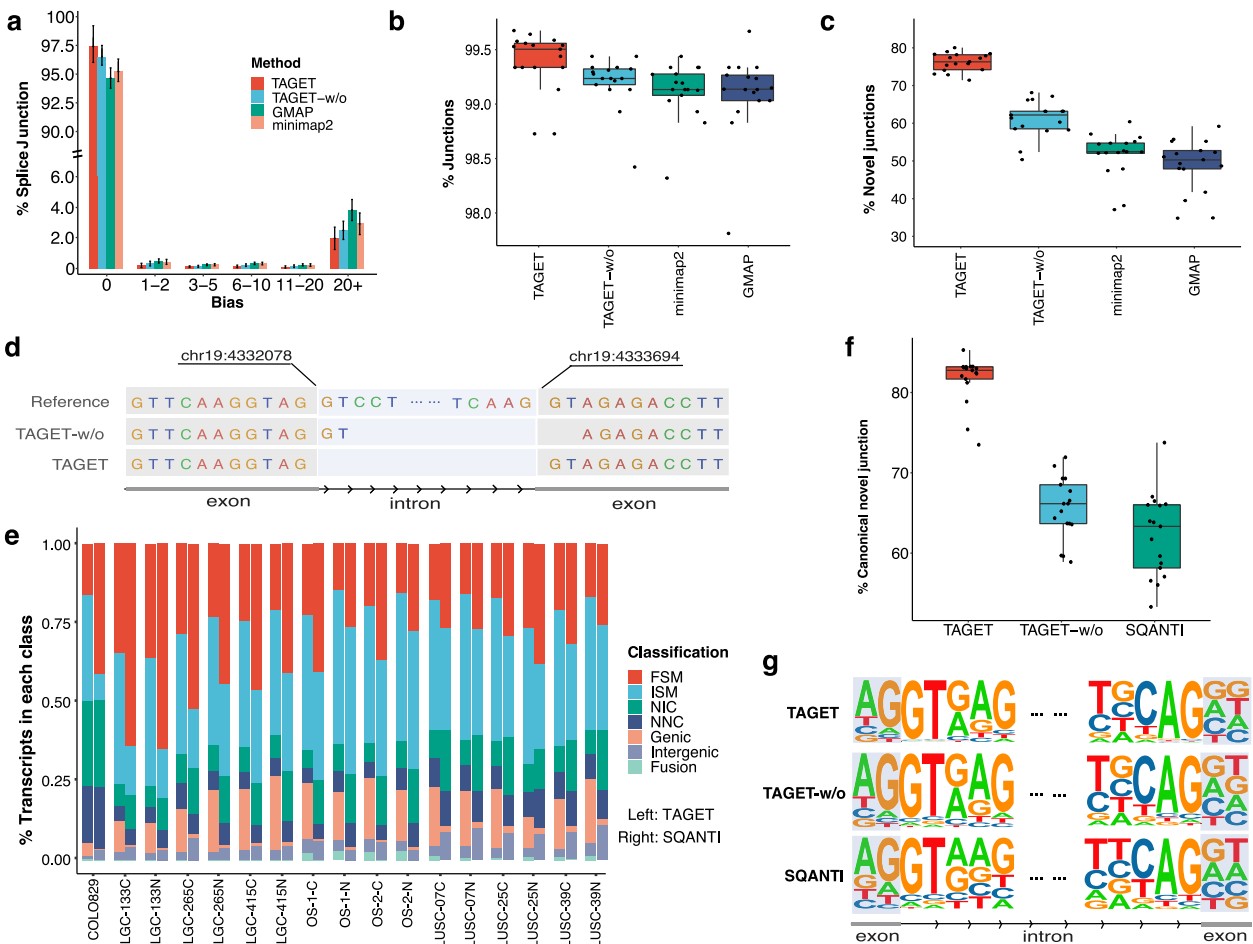

**Fig. 2 | Performance evaluation on mapping and annotation. a** The distribution of splice site biases of TAGET, TAGET-w/o (TAGET without CNN), GMAP, and minimap2 (*n* = 17 samples). The height of the bars refers to the average percentages of splice junctions. Error bars represent the standard deviations. **b, c** The proportions of all (**b**) and novel (**c**) splice junctions having at least two RNA-seq supporting reads or in the reference database (*n* = 17 samples; junctions supported by multiple reads were counted multiple times). The center line in the boxplot is the median, bounds of boxes are the interquartile of the data, whiskers represent minima/maxima excluding outliers and dots represents outliers of beyond 1.5*Interquartile Range (IQR) from either end of the box. **d** An example of improved mapping by the CNN correction. The first line: the reference genome; The second line: the mapping before the CNN correction; The third line: the mapping after the CNN correction. After the correction, the junction becomes a canonical junction. **e** Distribution of the seven transcript types annotated by TAGET and SQANTI, respectively. **f** The percentages of canonical junctions in novel junctions reported by TAGET, TAGET-w/o, and SQANTI (*n* = 17 samples). The center line in the boxplot is the median, bounds of boxes are the interquartile of the data, whiskers represent minima/maxima excluding outliers and dots represents outliers of beyond 1.5*Interquartile Range (IQR) from either end of the box. **g** Motif analysis of novel junctions in LGC-133C identified by TAGET (top), TAGET-w/o (middle) and SQANTI (bottom). Showing on the left is the 5' splice site; on the right is the 3' splice site. Both ends include 2 nucleotides of exons and 5 nucleotides of introns.

mapping algorithms, such as HISAT2[16] and STAR[17], can sensitively predict junction positions, but they try to maintain the best local alignments, which could cause splitting of exons into several parts (Supplementary Fig. 1b). To fully utilize the advantages of long-read and short-read mapping algorithms and avoid their shortcomings, TAGET combines mappings from both long-read and short-read algorithms to improve the alignment (Fig. 1b, "Methods" section). After the integrative mapping, we observe that many splice sites are still incorrectly mapped because of the mapping ambiguity. We thus further use a CNN model[18] for local alignment adjustment to improve splice junction prediction ("Methods" section).

After mapping, transcripts can overlap with multiple known isoforms in the reference transcriptome database. TAGET determines the best match of the isoforms in the reference database ("Methods" section) and classifies the annotated transcripts into seven classes: FSM (Full Splice Match), ISM (Incomplete Splice Match), NIC (Novel in Catalog), NNC (Novel Not in Catalog), Genic, Intergenic, and Fusion. TAGET can quantify gene expression as well as expressions of known and novel isoforms with the full-length transcript data ("Methods" section). The full-length transcript data allow discovering DIU genes in different conditions. TAGET creates a contingency table for each gene and performs Fisher's exact test to detect DIU genes ("Methods" section).

The public Iso-seq data used for assessing the performance of TAGET is a dataset sequenced from the cancer cell line COLO829. The newly sequenced data includes seven pairs of matched cancer and normal datasets. We also performed RNA-seq from these cancer and normal samples. Three of the seven pairs (LGC-133, LGC-415, and LGC-265) were from laryngocarcinoma (LGC) patients, three pairs (LUSC-07, LUSC-25, LUSC-39) were from lung squamous cell carcinoma patients, and the last pair was from a patient with osteosarcoma (OS). Two technical replicates (OS-1 and OS-2) were sequenced for both cancer and normal tissues of the OS patient. We obtained 16,000-170,000 transcripts with at least two CCS reads per sample (Supplementary Table 1 and Supplementary Fig. 2). We also evaluated the performance of TAGET on the ONT data using the data sequenced from the GM12878 cell line and from lung cancer cell lines[19].

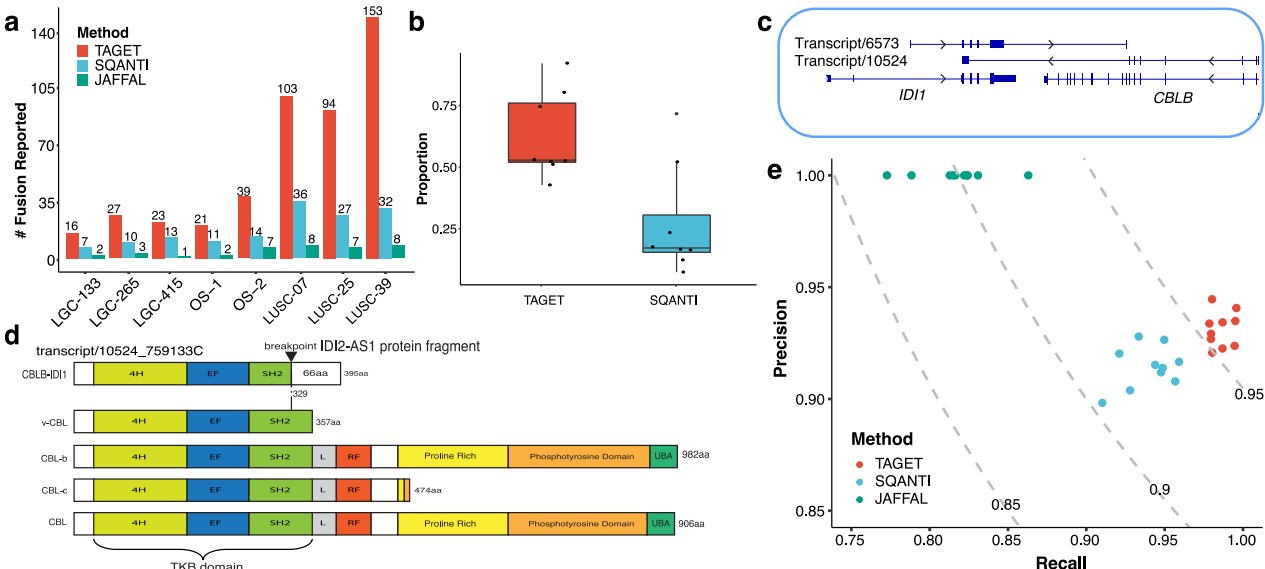

**Fig. 3 | Gene fusion discoveries. a** The numbers of reported somatic fusions by TAGET, SQANTI, and JAFFAL. **b** Proportions of gene fusions reported by TAGET and SQANTI that have at least two supporting short reads ($n = 8$ samples). The center line in the boxplot is the median, bounds of boxes are the interquartile of the data, whiskers represent minima/maxima excluding outliers and dots represents outliers of beyond 1.5*Interquartile Range (IQR) from either end of the box. **c** The *CBLB-IDI1* gene fusion. **d** The structure of the fusion *CBLB-IDI1* and the CLB proteins. **e** The gene fusion simulation results given by TAGET, SQANTI, and JAFFAL. The dashed lines are the contour lines with constant F-scores (F-scores are marked on the lines).

## TAGET enables accurate transcript mapping in Iso-seq data

We evaluated the transcript alignment accuracy of TAGET by comparing it with long-read alignment tools (minimap2, GMAP). To evaluate the effect of the CNN model, we also included the results of TAGET when the CNN model was turned off (TAGET-w/o). We found that TAGET mapped more reads than other algorithms in most cases (Supplementary Fig. 3a and Supplementary Data 1). The numbers of junctions reported by TAGET, TAGET-w/o, minimap2, and GMAP were very close (Supplementary Fig. 3b). We calculated distances between predicted junctions and junctions in the Ensemble database. Overall, 97.4% of predicted junctions were the same as the junctions in the Ensemble database. In comparison, 95.2% and 94.3% of junctions predicted by minimap2 and GMAP were the same as the junctions in the Ensemble database, respectively (Fig. 2a and Supplementary Data 2).

To further evaluate the accuracy of junction mapping, we aligned short reads of RNA-seq data to the reference genome. On average, 99.3% of splice junctions predicted by TAGET were supported by at least two short reads or in the reference database, slightly higher than those of minimap2 (99.1%) and GMAP (99.1%) (Fig. 2b and Supplementary Data 3). For novel junctions, 75.2% of junctions predicted by TAGET (range from 71.3% to 80.0%) were supported by at least two short reads, much higher than those given by minimap2 (mean 48.5%, range from 36.7% to 60.2%) and GMAP (mean 43.8%, range from 34.4% to 59.0%; Fig. 2c and Supplementary Data 3).

We further investigated the junction positions adjusted by the CNN model. Around 60% of predicted novel junctions given by TAGET-w/o can be supported by RNA-seq (Fig. 2c), much lower than that given by TAGET, indicating that the CNN model effectively improved the accuracy of splice site prediction. Figure 2d shows an example of the CNN adjustment. The transcript had a dinucleotide GT that can be mapped to either the position chr19:4332078 or chr19:4333694. Both mappings matched the reference genome without any error. Before the adjustment, TAGET reported the first mapping, but the splice junction was different from the known annotation and non-canonical, and thus likely incorrect. After the adjustment, TAGET reported the second mapping.

## TAGET provides accurate transcript annotation and splice junction prediction

We compare the annotations given by TAGET and the available state-of-the-art algorithm SQANTI[10]. Considering the accuracy of different mapping algorithms, we provided the mappings of minimap2 to SQANTI for annotation. TAGET and SQANTI gave overall similar gene annotations (Fig. 2e and Supplementary Fig. 3c). Large annotation differences include Genic, FSM, and ISM classes, which were mainly due to different definitions of these transcript classes in TAGET and SQANTI. For example, in the LGC-133C sample, among the 7241 transcripts that were annotated differently by SQANTI and TAGET, 5556 (76.7%) of the inconsistency can be attributed to the different definitions of Genic, FSM, and ISM transcripts.

TAGET annotated a larger proportion of transcripts as Genic transcripts. These transcripts were monoexonic transcripts whose transcriptional start or termination sites were inconsistent with the transcripts in the Ensemble database (Supplementary Fig. 3d). In our view, it was more appropriate to classify them as Genic transcripts than as FSM/ISM/NIC/NNC transcripts. In terms of FSM and ISM, SQANTI annotates a transcript as FSM if all of its splice junctions match a reference isoform. TAGET further requires that FSMs match the transcription start sites (TSS) and the polyadenylation sites (PAS) to reference isoforms, and do not have intron retentions. Hence, TAGET annotated fewer FSMs and more ISMs than SQANTI. Supplementary Figure 3e shows such an example. The transcript 10006 was annotated as FSM by SQANTI and ISM by TAGET. Although every splice junction of this transcript matches the reference isoform STX7-202, the first exon of STX7-202 was not fully covered by transcript 10006. Thus, classifying transcript 10006 as an ISM would be more appropriate. Supplementary Figure 3f shows another example, in which transcript 10139 was annotated as FSM by SQANTI and NIC by TAGET.

We then compared the accuracy of splice site prediction. The numbers of splice junctions reported by TAGET and SQANTI were very similar (Supplementary Fig. 4). On average, 99.7% and 99.8% of known junctions given by TAGET and SQANTI were canonical junctions, respectively (Supplementary Fig. 3g and Supplementary Data 4). Note that, for known splice junctions, SQANTI always reported the junction positions in the reference database even if there were mismatches.

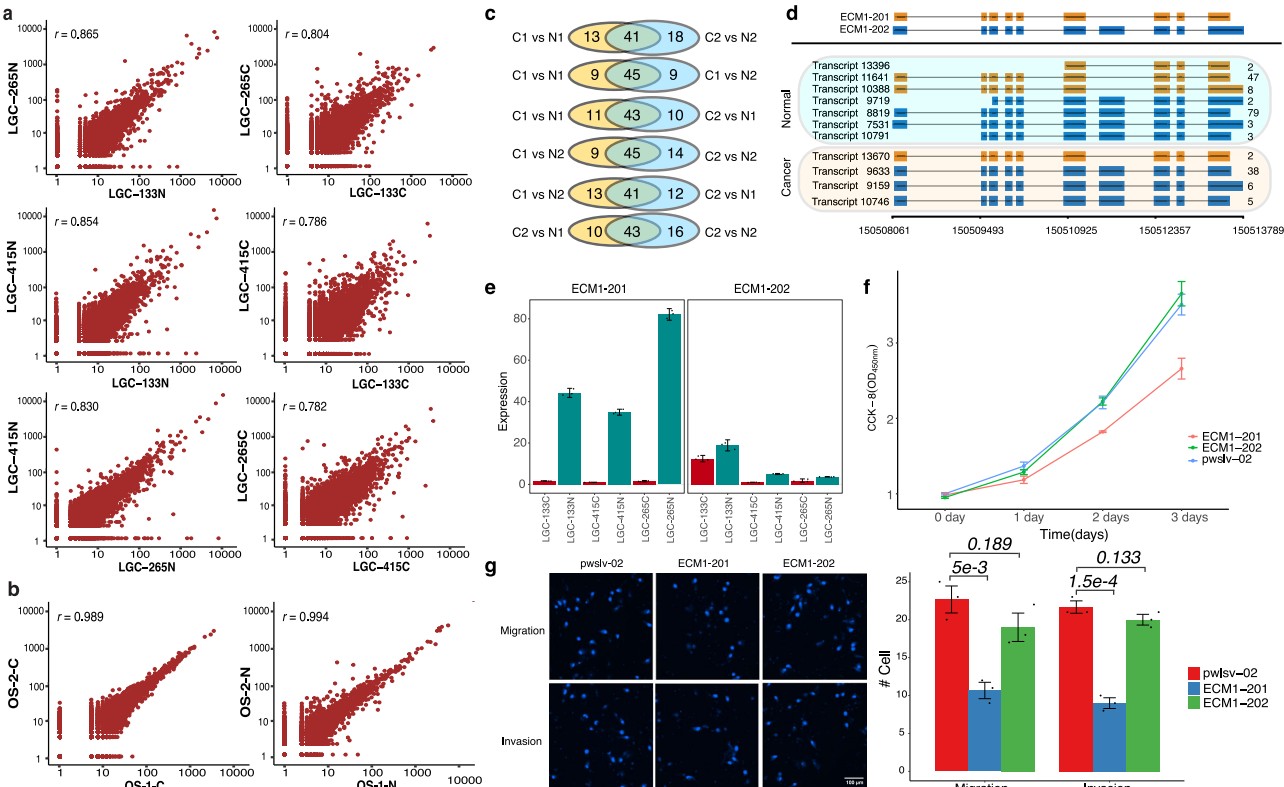

**Fig. 4 | TAGET gives accurate gene expression quantification and identifies *ECM1* as an important DIU gene. a** Scatter plots of Iso-seq expressions between pairs of LGC normal/tumor samples. **b** Scatter plots of Iso-seq expressions between replicates of OS normal/tumor samples. **c** The Venn plots of numbers of DIU genes reported by TAGET. C1: OS-1-C; C2: OS-2-C; N1: OS-1-N; N2: OS-2-N. C1 vs N1 means the DIU genes detected by comparing OS-1-C with OS-1-N. Similar for the others. **d** *ECM1* Isoforms. Top panel: the two isoforms in the reference database. Bottom panel: the detected isoforms in LGC-133 tumor and normal samples. The numbers at the right side indicate the CCS read number of each transcript. **e** The comparison of expression levels of ECM1-201 and ECM1-202 (*n* = 3 experiments). The height of the bars refers to the average expressions of triplicate PCR experiments. Error bars represent the standard deviation of the triplicate PCR experiments. **f** CCK-8 assays validating the effect of ECM1-201 and ECM1-202 overexpressing on 6-10B cell proliferation (*n* = 3 experiments, two-sided Student's *t* test). The center of the bars refers to the average cell counts. Error bars represent the standard deviations. *P* values: 0.027 (1 day; ECM1-201 vs. pwslv-02); 0.018 (1 day; ECM1-202 vs. pwslv-02); 0.0014 (2 days; ECM1-201 vs. pwslv-02); 0.0018 (2 days; ECM1-202 vs. pwslv-02); 0.037 (3 days; ECM1-201 vs. pwslv-02); and 0.0021 (3 days; ECM1-202 vs. pwslv-02); **g** Transwell assays validating the effect of ECM1-201 and ECM1-202 on 6-10B cell migration and invasion. The height of the bars refers to the average expressions. Error bars represent the standard deviation of three independent experiments (*n* = 3 experiments; two-sided Student's *t* test).

Hence, the proportion of canonical junctions of SQANTI was slightly higher than that of TAGET. For novel junctions, on average, 82.3% junctions reported by TAGET were canonical, much higher than those of SQANTI and TAGET-w/o (Fig. 2f, Supplementary Data 5). Motif analysis also confirmed the enrichment of GT and AG at the 5' donor splice sites and the 3' acceptor sites, respectively (Fig. 2g). We also found that, on average, 97.4% of splice junctions (ranging from 94.1% to 98.4%) predicted by SQANTI were supported by at least two short reads, slightly lower than the junctions predicted by TAGET (Supplementary Fig. 3h and Supplementary Table 2). For novel junctions, 54.8% of junctions (ranging from 40.5% to 63.4%) predicted by SQANTI were supported by at least two short reads, much lower than the junctions given by TAGET (Supplementary Fig. 3i, Supplementary Table 3).

To further confirm the accuracy of TAGET's prediction of novel isoforms, we randomly selected 73 NIC/NNC isoforms predicted from COLO829, LGC-133C, LUSC-07C, and LUSC-39C for validation, and 63 (86.3%) were validated by Sanger sequencing (Supplementary Data 6-10).

We also performed simulation to evaluate the sensitivity and precision of TAGET and compared with other methods, including SQANTI[20], IsoQuant[20], StringTie2[21] and FLAIR[22]. Briefly, we randomly deleted annotations of X% (X = 5, 10, 20, 30, and 50) of genes and evaluated sensitivity as the proportion of deleted junctions that could

be recovered, and precision as the proportion of true junctions in the discovered junctions (Methods). We found that TAGET and SQANTI achieved much higher sensitivities and slightly lower precisions than StringTie2 and FLAIR (Supplementary Fig 5).

## TAGET accurately detects gene fusions with high sensitivity

TAGET uses full-length transcript data to detect gene fusions. TAGET reported more fusions than SQANTI and JAFFAL[23] (Fig. 3a). For gene fusions predicted by TAGET, on average, 63% (ranging from 44% to 92%) of the fusion junctions were supported by at least 2 reads from RNA-seq data. In comparison, only 29% (ranging from 10% to 72%) of the fusion junctions predicted by SQANTI were supported by RNA-seq data (Fig. 3b). We then randomly selected 35 TAGET-predicted gene fusions for validation. Among these gene fusions, 30 gene fusions (85.7%) were validated by Sanger sequencing (Supplementary Fig. 6 and Supplementary Data 11–14).

A number of fusions detected by TAGET involve known cancer genes such as *HNRNPA3-NFE2L2*, *SNRNP25-TSC2*, *CCDC83-PICALM*, and *CBLB-IDI1*. We observed two transcripts involving *CBLB* and *IDI1* in sample LGC-133C (Fig. 3c). These two transcripts are on opposite strands of the genome. The Sanger-validated *CBLB-IDI1* transcript (transcript 10524 in Fig. 3c) corresponds to exons 1–7 of *CBLB* (CBLB-201) and one exon from *IDI1*, indicating a fusion between *CBLB* and *IDI1*. On the other hand, most exons of the other transcript *IDI1-CBLB*

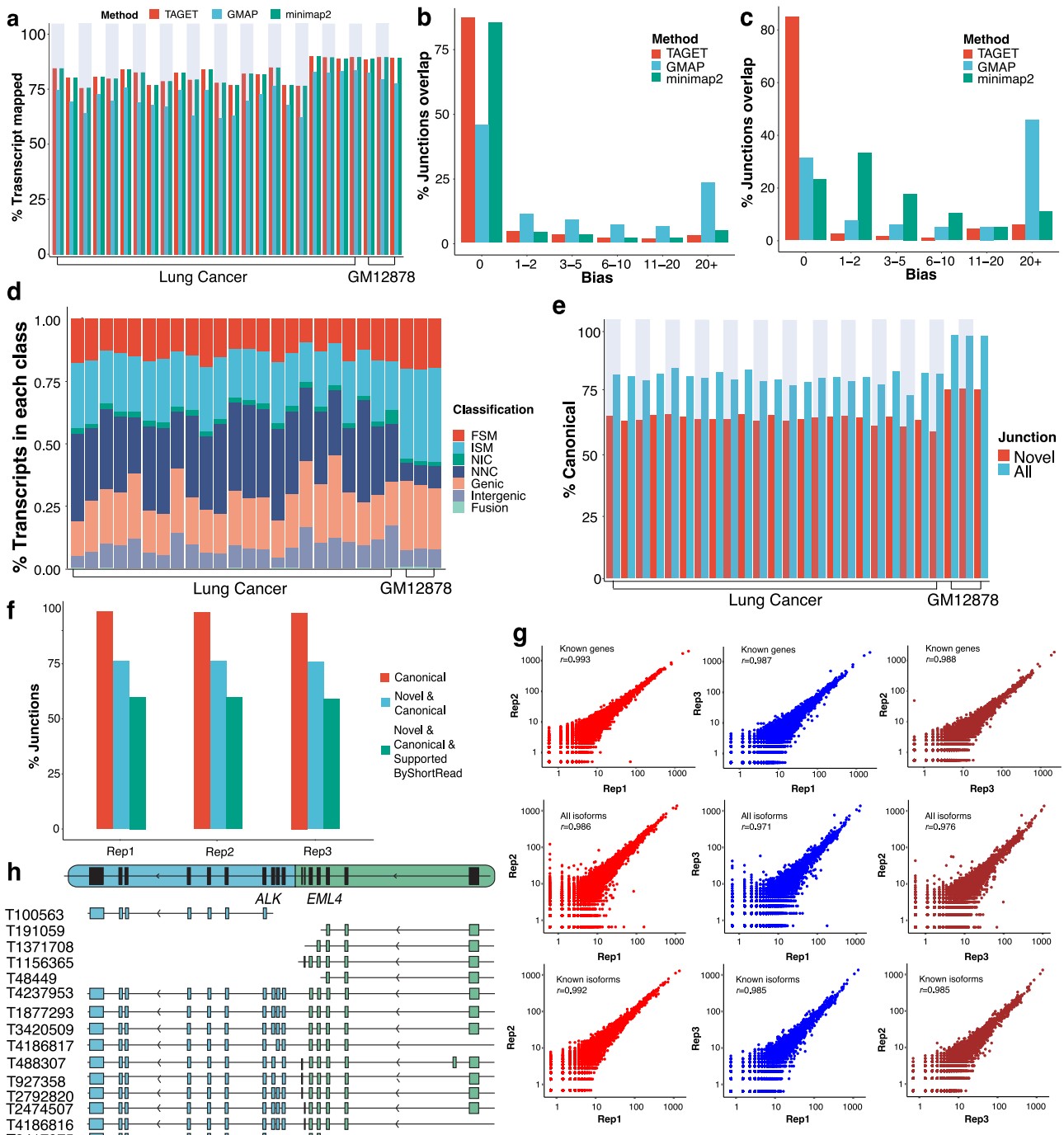

**Fig. 5 | Performance of TAGET on Nanopore datasets. a** Proportions of mapped long-reads by TAGET, GMAP, and minimap2. **b**, **c** The distribution of splice site biases of TAGET, GMAP, and minimap2 for GM12878 (**b**) and lung cancer (**c**). **d** The percentage of seven types of annotations given by TAGET. **e** The percentages of canonical junctions in novel junctions and all junctions. **f** The proportions canonical junctions (red), novel canonical junctions (blue), and novel canonical junctions supported by at least two short reads (green) in the three replicates of GM12878. **g** The consistency of expression of known genes (the top line), all isoforms (the middle line), and known isoforms (the bottom line) in three replicates of GM12878. **h** The *EML4-ALK* gene fusion.

(transcript 6573 in Fig. 3c) match those of *IDI1*, and only its last exon matches the intronic sequence of *CBLB*. The two transcripts are possibly generated by an inter-chromosomal translocation. We did not observe expression of the known isoforms of *CBLB* and only the fusion *CBLB-IDI1* was expressed in the sample LGC-133C.

CBL proteins including CBL, CBLB, and CBLC are E3 ubiquitin-protein ligases that can promote degradation of receptor tyrosine kinases such as EGFR[24,25]. CBL proteins (other than vCBL) have a N-terminal tyrosine-kinase-binding (TKB) domain and a RING finger

(RF) domain. The truncated CBL protein vCBL lacks a RF domain and can induce leukemia and lymphoma in mice[26]. The RF domain and the α-helix linker between the TKB and RF domain are often mutated or deleted in oncogenic CBL variants[27], which underscores the importance of these domains for the proper function of CBL proteins. The *CBLB-IDI1* fusion retains most sequences of the TKB domain but completely lost its RF domain and ubiquitin binding domain. This results in a vCBL-like truncated protein (Fig. 3d). Therefore, *CBLB-IDI1* could potentially represent an oncogenic variant of *CBLB*. Unlike other

known oncogenic variants, this potential oncogenic variant is generated from a gene fusion instead of mutations or deletions.

We further did simulation studies to compare the performance of TAGET with SQANTI and JAFFAL. We generated ten simulated Iso-seq datasets, each of which had 300 gene fusions at various expression levels ("Methods" section). Overall, TAGET had the best performance in terms of sensitivity and F-score (Fig. 3e). JAFFAL applied various filters to filter potential false positives. While these filters successfully eliminate many incorrect predictions, they also result in the removal of some true discoveries. So, JAFFAL had a higher specificity but a lower sensitivity than TAGET and SQANTI.

### TAGET provides accurate gene expression quantification and identifies *ECM1* as a DIU gene

We compared the expression given by Iso-seq and RNA-seq and found that they were positively correlated (Spearman's correlation -0.5, Supplementary Fig. 7a). We also compared the expression given by LIQA[28], a recently developed long-read expression quantification tool, and found the expression reported by TAGET had higher correlation with RNA-seq data than that of LIQA (Supplementary Fig. 7b). We performed DEG analysis between three pairs of LGC samples using Iso-seq and RNA-seq. Gene Ontology (GO) analyses of the DEGs given by Iso-seq and RNA-seq gave very similar enrichment of GO terms (Supplementary Fig. 8). The pairwise Iso-seq isoform expression correlations between the normal (or cancer) samples were high (Pearson's correlation -0.8, Fig. 4a, Supplementary Fig. 9, and Supplementary Data 15 and 16). The correlations of Iso-seq isoform expressions between the two technical replicates of the OS patient were even higher (0.989 for OS-C and 0.994 for OS-N, Fig. 4b and Supplementary Data 17). This demonstrates the high reproducibility of Iso-seq transcript analysis. In the DIU analysis, we identified 53-59 DIU genes between the OS cancer and normal samples, with a considerable overlap of DIU genes between the comparisons (Fig. 4c and Supplementary Data 18). No DIU genes were reported between pairs of the OS technical replicates, indicating the high reproducibility of the DIU analysis. For example, the DIU analyses identified 54 DIU genes between OS-1-C and OS-1-N and 59 DIU genes between OS-2-C and OS-2-N, among which 41 DIU genes were common (Fig. 4c).

We further performed DIU analysis for the three LGC samples (false discovery rate or FDR < 0.05, Supplementary Data 19–21). *ECM1* was one of five common DIU genes in all LGC samples. *ECM1* had two main isoforms: ECM1a (ECM1-202) and ECM1b (ECM1-201; Fig. 4d). PCR validation showed that the expression of ECM1b was significantly depleted in cancer samples, while ECM1a had a similar expression level in both cancer and normal samples, leading to an overall depletion of *ECM1* in cancer samples (Fig. 4e and Supplementary Data 22). RNA-seq data showed that *ECM1* was downregulated in cancer samples and was a DIU gene in three LGC cancer samples (Supplementary Fig. 10).

*ECM1* was known to stimulate the proliferation of endothelial cells, promote angiogenesis, and play oncogenic roles in cancer development in various cancer types[29–32]. Contrary to the oncogenic role of ECM1a, recent researches showed that ECM1b acted as a cancer suppressor[33]. To validate the tumor-suppressive role of ECM1b, we constructed two plasmids containing either ECM1a or ECM1b and overexpressed these two isoforms together with a pwslv-02 control plasmid in the 6-10B cell line (a nasopharyngeal carcinoma cell line). Cell counting K-8 (CCK8) assay showed that two days after transfection, cells transfected with ECM1b began to display significantly decreased proliferation than cells transfected with ECM1a or the pwslv-02 control (Fig. 4f and Supplementary Data 23). Cells transfected with ECM1a did not show a significant difference in proliferation compared with the pwslv-02 control cells. We also evaluated the migration/invasion capability of 6-10B cell line using the transwell assay. Our results showed that ECM1b strongly inhibited both cell migration and invasion by 50%, but ECM1a did not show significant effect (Fig. 4g and

Supplementary Fig 11a). To further validate our findings, we overexpressed ECM1a and ECM1b in another three cell lines: 5-8F (nasopharyngeal carcinoma), HeyA8 (ovarian cancer) and AU565 (breast cancer) (Supplementary Fig 11). Consistent with the result in 6-10B cells, ECM1b showed an inhibitory effect in all these cell lines. The role of ECM1a in cell migration and invasion varied across cancer types: it had no impact on the 5-8F cells, but showed significant promoting effect on HeyA8 and AU565 cells, consistent with the finding in a recent study[34]. These results suggested different roles of the two *ECM1* isoforms in various cancer types, consistent with our finding that *ECM1* was a DIU gene between tumor and normal tissues in the three LGC samples.

### TAGET provides accurate alignment, annotation, and expression quantification for ONT data

In this section, we evaluate the performance of TAGET on ONT data. We considered a GM12878 cell line with three replicates[35] and 23 samples from lung cancer cell lines[19]. The percentages of mapped reads were largely similar among TAGET, GMAP and minimap2 (Fig. 5a), but splice junctions given by TAGET were more accurate, especially for the lung cancer data (Fig. 5b, c). Overall, due to the higher error rate of ONT data, their mappings were less accurate than Iso-seq data. We annotated these transcripts and found that lung cancer cell lines had increased proportions of NNC and fusion transcripts than normal cells (Fig. 5d). Most novel junctions of the annotated transcripts were canonical or supported by short reads (Fig. 5e, f).

We then compared the gene and isoform expression between pairs of the three GM12878 replicates. The gene and transcript expressions of three replicates were highly correlated (e.g., Pearson's correlation of all detected isoforms >0.971; Fig. 5g). Besides, both the DEG and DIU analyses between GM12878 replicates did not yield significant results (FDR > 0.05), indicating that there was no bias introduced during our analysis. To further confirm the accuracy of gene expression quantification, we calculated the Pearson's correlations between expressions given by TAGET and LIQA, and found that they were highly correlated (-0.83; Supplementary Table 4).

Lastly, we investigated gene fusions detected in the lung cancer cell lines. In the cell line H2228, TAGET detected the well-known fusion *EML4-ALK* (Fig. 5h). Interestingly, we observed that this fusion had 8 different isoforms, which was difficult to be detected by RNA-seq data. The breakpoints of transcript 2474507 and 4186816 were consistent with the known *EML4-ALK* fusion in the CCLE database. All other breakpoints were novel. Moreover, some transcripts share the same breakpoints but corresponded to different isoforms. For example, the transcripts 488307 had the same breakpoints as transcript 927358 and 2792820, but transcript 488307 had one extra exon on the *EML4* end of the fusion. This high diversity level of the *EML4-ALK* fusion might confer cancer cells additional selective advantages. The functions and impacts of these fusion transcripts are unknown and need to be further studied.

## Discussion

The Iso-seq technology has tremendous potential for various transcriptome analyses. Here, we developed TAGET as a unified computational framework that can analyze Iso-seq data from alignment to downstream quantification analyses. TAGET provides a convenient toolbox for analyzing Iso-seq data of different organisms using the corresponding reference genome and transcript annotation files. One major improvement is that TAGET provides more accurate transcript alignment and junction prediction, which is critical for transcriptome studies such as alternative splicing and novel isoform discovery. TAGET improve alignments by integrating available long-read and short-read mappings and by local fine adjustment of junctions using a CNN model.

With its long-read, isoform expression could be more readily obtained from Iso-seq than RNA-seq. In addition, unlike RNA-seq, Iso-seq does not involve PCR amplification and do not have PCR biases. Thus, Iso-seq could provide more accurate gene expression. However, partly because of the lack of computational tools, these advantages of Iso-seq are not fully utilized in the literature and gene expressions are often still estimated from RNA-seq in studies having Iso-seq data[36–38]. TAGET provides functionalities of expression analyses commonly used in RNA-seq as well as the DIU analysis tool that is difficult for RNA-seq data. With this DIU tool, we successfully detected a DIU gene *ECM1* between tumor and normal samples and showed that the two isoforms of *ECM1* might play different roles in tumor.

One limitation of Iso-seq expression is that the number of detected genes by Iso-seq is much less than that by RNA-seq. This may be due to two possible reasons. First, Iso-seq uses CCS reads to improve sequencing quality. Single nucleotides are covered and sequenced several times and the effective coverage is substantially reduced. Second, the Iso-seq preprocessing pipeline performs a clustering analysis of the CCS reads and it is suggested that clusters with only one CCS read should be filtered because of their lower quality. This step filters about 80% of the CCS reads. Many of these filtered reads come from true expressed isoforms. More advanced computational tools that can properly use these CCS reads should be able to significantly increase the detected genes by Iso-seq while maintaining its low false discovery rate.

TAGET can be viewed as a transcript alignment tool. TAGET improves full-length transcriptomics data analyses by optimizing the alignment and the annotation. In comparison, recently developed computational tools, such as ESPRESSO[39], FLAIR, StringTie2, and Iso-Quant, use transcript reconstruction to improve transcript analyses of long-read data. These reconstruction methods utilize multiple reads for transcript detection and thus have higher precisions. TAGET and SQANTI analyze long-reads independently and generally have higher sensitivity. PacBio's HiFi reads have very high precision. The transcript reconstruction can slightly improve the precision of transcript analyses of HiFi data, but also lead to decreased sensitivity. We believe that TAGET is more suitable for the high-quality Iso-seq data and the reconstruction methods are more suitable for error-prone long-read data. Further, compared with these transcript reconstruction tools, TAGET could also perform annotation and expression quantification analyses, and may have wider applications.

Overall, our results strongly validated the performance of TAGET on Iso-seq data. This tool is especially appealing in the light of recent findings showing that utilizing isoform-level expression data may also lead to more reliable gene-level expression estimates[40]. The long-read sequencing technology continues to improve in terms of the accuracy and number of reads and genes captured. As data quality increases, we expect that TAGET will have significant potential in third-generation sequencing data analysis and isoform-based research.

## Methods

### Ethical regulations
The study was conducted in accordance with the ethical standards of the Research Ethics Committee of Harbin Medical University Cancer Hospital with patient's informed content. (Approval No: JJ2022LH1305)

### Alignments of full-length transcript data
Polished transcripts are first mapped to the reference genome by minimap2[9] (version 2.24) or GMAP[15] (version 2017-11-15) (by default minimap2). TAGET then scans the long-read using sliding windows, takes all the subsequences (short-reads) in the sliding windows and maps the short-reads to the reference genome by HISAT2[41] (version 2.2.1) or STAR[17] (version 2.7.8a). Then, TAGET assembles long-read and short-read mappings to get a more accurate mapping of the polished long-read.

More specifically, let $S = s_1 \cdots s_L$ be a polished long read of length $L$. TAGET scans the long-read with sliding windows of length $h$ (by default $h = 250$) and step size $t$ (by default $t = 100$), takes the subsequences (called short-reads) in the windows and aligns them with HISAT2 or STAR. In total, there are $n = \lceil \frac{L-h+t}{t} \rceil$ such short reads, $R_1 = s_1 \cdots s_h$, $R_2 = s_{1+t} s_{2+t} \cdots s_{h+t}, \cdots, R_n = s_{L-h+1} \cdots s_L$. Using simulation data, TAGET achieves its best final alignment with $h = 250$ and $t = 100$ and thus uses these values as the default. For the long-read S, we then have one long-read mapping and $n$ short-read mappings, each nucleotide of S can have at most $n + 1$ possible mapping positions in the reference genome. If a nucleotide has only one possible mapping position (all its long-read mapping and short-read mapping positions are the same), the nucleotide is uniquely mapped; Otherwise, the nucleotide is multiply mapped. A long-read is thus partitioned into blocks of continuously uniquely mapped nucleotides and continuously multiply-mapped nucleotides. For uniquely-mapped blocks or multiply-mapped blocks of no more than 10 bp, TAGET takes the mapping positions given by the long-read mapping algorithm as their mappings. For multiply-mapped blocks of more than 10 bp, TAGET uses the following procedure to determine their mapping positions.

Given a block of continuously multiply-mapped nucleotides, $B = b_1 b_2 \cdots b_l$. Suppose that the $i$ th nucleotide $b_i$ has $n_i$ possible mapping positions $p_{i1} < p_{i2} < \cdots < p_{in_i}$ (if these positions are at different chromosomes, they are first sorted alphabetically by chromosome names and then by their coordinates) and TAGET assigns each mapping position a mapping quality score $\omega_{ij} = 1/n_i, j = 1 \ldots n_i$. Given an alignment $A$ of the block $B$, define $D_1(A)$ and $D_2(A)$ as the distances to the upstream and downstream of the block $B$. If $B$ is mapped to a different chromosome by the alignment $A$ from its upstream or downstream mapping, $D_1(A)$ or $D_2(A)$ will be given a large value (default 1000). Denote $\mu_1, \mu_2, \mu_3, \mu_4$ (by default $\mu_1 = 2$, $\mu_2 = 2$, $\mu_3 = 1$, and $\mu_4 = 1$,) as tuning parameters. We use the following "Algorithm 1" to align the block $B$.

**Algorithm 1. Data**: the mapping positions $p_{ij}$ and mapping quality scores $\omega_{ij}$ ($i = 1, \cdots, l, j = 1, \cdots, n_i$)

> **Result**: The best local alignment combination
> *Alignments* <- empty list; $\tau = 1$;
> While not all $p_{ij}$ in *Alignments* do
> > Set the local alignment $A_\tau = p_{ij_i}$, where $p_{ij_i}$ = the first $p_{ij}$ not in *Alignments*;
> > Set $k = i$;
> > **while** $k < l$ **do**
> > > suppose $A_\tau = p_{ij_i}, p_{i+1_{j_{i+1}}}, ..., p_{kj_k}$
> > > **if** there exists $j_{k+1} = 1, \cdots, n_{k+1}$ such that $p_{kj_{k+1}}$ and $p_{kj_k}$ at the same chromosome and $p_{kj_{k+1}} = p_{kj_k} + 1$, **then**
> > > > Set $A_\tau = p_{ij_i}, p_{i+1_{j_{i+1}}}, ..., p_{kj_k}, p_{kj_{k+1}}$
> > > **else**
> > > > **break**
> > > Set $k = k + 1$
> > **end**
> > Add $A_\tau$ into *Alignments*, set its quality score as $Q_\tau = \sum_{h=i}^{length} \omega_{hj_h}$
> > Set $\tau = \tau + 1$
> **End**

Find the alignment combination $A = A_{i_1}, A_{i_2}, \ldots, A_{i_k}$ ($i_1 < i_2 < \cdots < i_k$) such that $\sum_{h=1}^{k} Q_{i_h} - \mu_1 O - \mu_2 k - \mu_3 D_1(A) - \mu_4 D_2(A)$ is maximized over all combinations in *Alignments*, where $O$ is the total pairwise overlaps among the alignments $A_{i_1}, \cdots, A_{i_k}$.

### Splice site correction using Convolutional Neural Network
The CNN model[18] for the local adjustment of the splice sites consists of three blocks. Each block consists of a convolutional layer, a pooling layer by the rectified linear units (ReLU) and a batch-normalization layer. The inputs of the CNN model are 21 bp sequences near splice

sites (both exon and intron sequences) and the outputs are the splice positions of the sequences. For each nucleotide of the input sequence, we take its 10 bp upstream and 10 bp downstream sequences. If a nucleotide does not have enough nucleotides upstream or downstream, we pad with Ns and get the 21 bp sequence. These 21 bp sequences are then encoded by converting A, C, G, T, N to 4-vectors (1,0,0,0), (0,1,0,0), (0,0,1,0), (0,0,0,1), (0,0,0,0). The convolution kernels are chosen as 3×3 kernels with 50 channels. The training data and the validation data are taken as 21 bp sequences around the splice sites listed in the Ensemble database ($N$ bp from exon and $21 - N$ bp from intron with $N$ randomly generated from 1 to 10). 90% of the sequences sampled from the Ensemble database are chosen as the training data and the remaining 10% of the sequences are the validation data. The loss function is chosen as the categorical cross-entropy loss and the Adam optimizer is used to minimize the loss function. The model is trained for 1000 epochs on NVIDIA GeForce GTX 1080 Ti GPUs. The learning rate of the optimizer is set to 0.01. This procedure is repeated 10 times to make the model more accurate. We find that the CNN model achieves an accuracy of 98% in the validation data (Supplementary Fig. 12).

## Transcript annotation

Following the definition in SQANTI[10] (version 7.4.0), TAGET classifies the annotated transcripts into seven classes. FSM (Full Splice Match) and ISM (Incomplete Splice Match) transcripts are transcripts that match reference transcripts at all splice junctions and at consecutive but not all splice junctions, respectively. NICs (Novel in Catalog) are transcripts with at least one novel splice junction formed from existing donors and acceptors or one novel combination of existing splice junctions. NNCs (Novel Not in Catalog) are transcripts containing at least one novel donor or acceptor. Intergenic novel transcripts lie outside any annotated genes. Genic novel transcripts are monoexonic transcripts that do not match any reference and lie within an annotated gene. Fusions refer to transcripts that contain at least two already annotated genes.

More specifically, given a transcript (long read) $T$, if its mapping given by TAGET does not overlap with any isoform in the reference database, TAGET will annotate it as an intergenic isoform. If the mapping of the transcript $T$ overlap with any isoform in the reference database, TAGET uses the following procedure to annotate $T$. Let $I$ be an isoform in the reference database that overlaps with $T$. Denote $L_{TI}$ as the overlap length of $T$ and $I$, $L_T$ as the length of the transcript $T$ that was not covered by the isoform $I$ and $L_I$ as the length of the isoform $I$ that was not covered by the transcript $T$. We define a score as $L_{TI} - \theta_1 L_I - \theta_2 L_T$ to measure the overlap of the transcript and isoform, where $\theta_1$ and $\theta_2$ are the penalty parameters (by default $\theta_1 = 0.5$ and $\theta_2 = 0.5$). TAGET considers all isoforms in the reference database that overlap with $T$ and chooses the one with the highest score as the candidate annotation of $T$ (denoted as $I_T$).

If all of the splice junctions of a transcript $T$ are consistent with either all or consecutive junctions from the candidate isoform $I_T$, it will be considered as a known transcript. A known transcript $T$ will be regarded as FSM if all of its splice junctions are consistent with those of its candidate isoform $I_T$, and its 3' and 5' ends are within 100 bp from the ends of its candidate isoform $I_T$. Otherwise, the known transcript will be classified as ISM. Novel transcripts are transcripts that have novel combinations of known splice junctions or novel splice junctions compared with the isoforms in the reference database. If a novel transcript $T$ has only known splice junctions of the reference genes and harbors their novel combinations, this transcript will be classified as NIC. If a novel transcript has at least one novel junction and at least one of its novel junctions unambiguously locates at two different genes and the subsequences mapped to either of the two genes are greater than 20 bp, the transcript is classified as a fusion. All other novel transcripts will be considered as NNC. For a monoexonic

transcript, if both of its ends are consistent with a reference monoexonic isoform, it will be considered as FSM; If the two ends of the monoexonic transcript are consistent with the splice sites of an exon of a multi-exonic reference isoform, this isoform will be considered as ISM; Otherwise, the monoexonic transcript will be classified as a genic novel transcript.

## Quantification and analysis of isoform expression

For each known isoform in the reference database, TAGET uses the number of FSM and ISM transcripts that are uniquely annotated to the known isoform as its observed transcript count. An FSM or ISM transcript is said to be uniquely annotated if its splice junctions match consecutive splice junctions of only one known isoform. If an FSM/ISM transcript is annotated to more than one known isoform, this transcript is not considered in the expression analysis. For novel isoforms, TAGET clusters the novel isoforms according to their splice junctions and uses the cluster size as their observed transcript count. Two isoforms in the same class (i.e., NIC or NNC) are clustered together if splice junctions of one isoform match consecutive splice junctions of the other isoform. The observed transcript count of a known gene is taken as the sum of the observed transcript counts of all its known isoforms. The expression value of a gene or an isoform is taken as the observed transcript count × 100,000 divided by the total transcript count of a sample (called transcript per hundred thousand or TPT).

TAGET applies the DEGseq[42] algorithm to genes' TPT expression values for DEG analysis and applies an exact two-sided binomial test from the R package statmod to the observed transcript counts for differential isoform expression analysis. To find DIU genes, TAGET creates a contingency table for each gene with rows as the isoforms of the gene and columns as the two conditions. Genes with very low expressions are excluded. Specifically, genes should satisfy the following conditions:

$$\sum_{i=1}^{m} E_{1i} \geq Q \text{ and } \sum_{i=1}^{m} E_{2i} \geq Q, \quad (1)$$

and for $\forall i$,

$$E_{1i} + E_{2i} \geq Q \text{ and } \frac{E_{1i} + E_{2i}}{\sum_{i=1}^{m} E_{1i} + \sum_{i=1}^{m} E_{2i}} \geq P, \quad (2)$$

where $E_{1i}$ and $E_{2i}$ are the observed transcript TPT value of the $i$ th isoform of the gene under two conditions, and $m$ is the number of isoforms of this gene. The $Q$ and $P$ are set to 4 and 5% by default. TAGET then performs Fisher's exact test to assign a $p$-value for each gene and adjusts the p-values using the FDR method.

## PacBio long-read sequencing

Total RNA samples were obtained from tumor and normal tissues of LGC and OS patients and PacBio sequencing was applied. cDNA was synthesized using the Clontech SMARTer PCR cDNA Synthesis Kit (634,925 and 634,926) and was then subjected to PCR amplification using PrimeSTAR® GXL DNA polymerase (R050A or R050B) with optimized cycles (typically 10-12). Amplified products of suitable sizes (0.5-6 Kb fragments) were further amplified by large-scale PCR to obtain enough DNA, and a SMRTbell Template Prep Kit (Pacific Biosciences) was used to construct a SMRTbell library. Sequel libraries with insertion fragments of 1–6 Kb were constructed and sequenced using a PacBio Sequel platform with V2.1 chemistry (Pacific Biosciences) and 10 hours movies.

## Transcriptome sequencing by RNA-seq

Total RNA was extracted from samples by the Trizol method. The sequencing library was obtained from the replicates using the NEB

Ultra RNA library Preparation Kit. The library preparations were subjected to paired-end sequencing using an Illumina Nova platform.

## Sensitivity and precision evaluation using simulation

We randomly deleted X% (X = 5, 10, 20, 30, and 50) of genes from the annotation file to evaluate the performance of different algorithms. All isoforms of the selected genes are removed and the obtained incomplete annotations are used as the reference data. We ran each tool using the incomplete annotation and evaluate the sensitivity and precision for junction detection. In our evaluation, positives are defined as the junctions of deleted genes, true positives (TP) as the reported junctions correctly recalled in the positive set, false positives (FP) as the reported junctions not in the positive set, and false negatives (FN) as the junctions in the positive set but not reported. Sensitivity and precision are defined as

$$\text{sensitivity} = \frac{TP}{TP + FN}, \tag{3}$$

$$\text{precision} = \frac{TP}{TP + FP}. \tag{4}$$

## Somatic gene fusion detection

Gene fusions were first detected in cancer samples and only candidates with at least one supporting CCS-read cluster of size two (i.e., the cluster contained at least two CCS reads) were kept as potential somatic fusion candidates. Then, we used Iso-seq data from matched normal samples to filter germline candidates. For a given somatic fusion candidate, if the matched normal sample had one or more CCS reads supporting the fusion, the candidate was removed. All the remaining candidates were somatic fusions. JAFFAL (version 2.3) was used in our study.

## Simulation for gene fusion detection

In the simulation study, we generated 10 different simulated datasets, each of which had 300 fusions and ~20,000 non-fusion transcripts. Fusion sequences were generated by randomly sampling partner genes, randomly positioning the breakpoints and concatenating the sequences from the sampled partner genes. Indels and SNPs were randomly added. Non-fusion transcripts were randomly sampled from GM12878, and transcripts matched the Cancer Cell Line Encyclopedia (CCLE) gene fusion datasets were removed. We also randomly assigned the number of supporting transcripts of each fusion ranging from 1 to 5.

## DEG analysis of RNA-seq data

Pair-end reads of cancer and normal samples were mapped to the reference genome (hg38) by using HISAT2. DESeq2 (version 3.15) was used for DEG analysis by comparing the paired cancer and normal samples. Gene ontology enrichment analysis (Biological process and KEGG) was generated by ShinyGO[43] with DEGs (Adjust p-value < 0.05), and protein-protein interaction networks functional enrichment analysis was generated by STRING[44].

## Validation of the novel isoforms and fusions by RT-PCR and Sanger Sequencing

Total RNA was extracted from specimens by the Trizol method[45,46]. Reverse transcription was performed using PrimeScript™ 1st Strand cDNA Synthesis Kit (6110 A, Takara). The PCR reactions were performed with 250 pg of template from COLO829 and LGC-133C. cDNA with KOD DNA polymerase (KMM-101, TOYOBO). PCR amplification was consisted of an initial cycle of 98 °C for 1 minute followed by 30 cycles of 98 °C for 10 seconds, 60 °C for 5 seconds and 68 °C for 10 seconds. The products were observed on a 2% agarose TAE gel. A

second PCR reaction was performed using a 1:1000 dilution of those PCR products and was carried out on a 2% gel. Appropriately sized bands were then excised and purified using a Qiagen gel purification kit to be sent for Sanger sequencing on an ABI 3730 Capillary Sequencer. See Supplementary Data 6 and 11 for the primer sequences used for PCR validation.

## Plasmid construction and cell transfection

The CDS sequences of ECM1-201 (ENST00000346569.6) and ECM1-202 (ENST00000346569.6) were synthesized by the GENEWIZ company and respectively assembled into pGL3 plasmid by Gibson method (Gibson Assembly® Master Mix, NEB). The human 6-10B cell lines were obtained from the Cell Resource Center, Peking Union Medical College, and cultured in RPMI-1640(Sigma-Aldrich, St. Louis, MO, USA) supplemented with 10% FBS (Life Technologies, Carlsbad, CA, USA) maintained at 37 °C in an incubator with 5% CO2. 6-10B cells were cultured in 6-well plates. When the cells grew to 50% density, the successfully constructed plasmids were transfected into cells respectively. The transfection was carried out by lipofectamine 2000 (Life Technologies, Carlsbad, CA, USA) following the manufacturer's protocols.

## Quantitative RT-PCR

RNA was extracted from tissues and cells using the Trizol method (Life Technologies, Carlsbad, CA, USA). RNA was reverse-transcribed into cDNA by reverse transcriptase (Takara, Dalian, China). The SYBR Green Mix (Vzayme, Nanjing, China) was used for qPCR reaction on ABI 7500 system. The expression of ECM1-201 and ECM1-202 was calculated using 2-ΔΔCT method, and GAPDH was used for internal reference. Primer sequences for RT-qPCR were listed in Supplementary Table 5.

## Cell proliferation

Cells were digested and centrifuged, $1 \times 10^4$ cells in each well were seeded into 96-well plates with 3 replicates in each group. After 24 hours, the cells were transfected and replaced with complete medium after 5 hours. After 0, 12, 24, 48, 72 hours, 10uL CCK reagent (Dojindo, Japan) was added to each well and cultured for 1 hour, and then the absorbance value at 450 nm was measured using the enzyme-labeled instrument (Multimode Reader, PerkinElmer, USA). All experiments were conducted in triplicate.

## Cell migration and invasion

Transwell plates (BD, USA) were used to verify cell migration ability. The cells were transfected 48 hours later. After digestion and centrifugation, $1 \times 10^4$ cells were added into the Transwell chamber (BD Biosciences). Serum-free medium was added into the upper chamber, and 20% FBS was added into the lower chamber. After 12 hours, the cells in the upper layer of the membrane were scraped with a cotton swab, fixed with 4% paraformaldehyde for 30 min, and stained with 0.1% DAPI for 10 min. Three fields were randomly selected from each plate and the number of cells in each field was counted under a fluorescence microscope (Ci, Nikon, Japan). The cell invasion assay was performed similarly to the migration assay, except that Matrigel (BD, USA) was coated to the upper chamber. All experiments were conducted in triplicate.

## Source of cell lines

Human cell lines COLO829, 6-10B, HeyA8, AU565, and 5-8F were purchased from the Center for Basic Medical Cells, Peking Union Medical College. The 6-10B, 5-8F, and HeyA8 cell lines were authenticated by STR DNA genotype analysis. Cell line Authentication Service was provided by Shanghai Yaji Biotechnology Co., Ltd. Cross-contaminations were not found in 6-10B, HeyA8, and 5-8F cell line.

## TAGET outputs

TAGET outputs the final corrected alignment results in BED format, and also reports all junctions of each read in a text file. TAGET outputs the read's annotations separately, so that junctions supported by multiple reads will be reported multiple times. TAGET also outputs the expression files of each gene and transcript, and a gene fusion file.

## Reporting summary

Further information on research design is available in the Nature Portfolio Reporting Summary linked to this article.

## Data availability

Datasets of COLO829 from the PacBio platform used in this study are available at https://www.zenodo.org/record/8319497. COLO829 101 bp Illumina short reads are available in NCBI database with an accession number SRR8615617. PacBio long-read sequencing and RNA-seq from laryngeal cancer patients, osteosarcoma, and lung squamous cell carcinoma patients generated in this study have been deposited in GSA database under accession code HRA002806. GM12878 datasets from the ONT platform are available in Gene Expression Omnibus (GEO) with an accession number GSE132766. Lung cancer cell lines from the ONT platform are available in DNA Data Bank of Japan (DDBJ) with an accession number DRA001859. The data HRA002806 in GSA are available under restricted access for privacy protection. Access for research purposes can be obtained by completing the application form via GSA. Users can register and login to GSA [https://ngdc.cncb.ac.cn/gsa-human/] and follow the guidance of "Request Data" [https://ngdc.cncb.ac.cn/gsa-human/document/GSA-Human_Request_Guide_for_Users_us.pdf] to request the data. Other data used in this study are publicly available. The authors declare that all data supporting the findings described in this manuscript are available in the article and its Supplementary Information files, and from the corresponding author upon request.

## Code availability

TAGET can be downloaded from https://github.com/XiDsLab/TAGET[47].

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

## Acknowledgements

This work was supported by the National Key R&D Program of China [2020YFE0204200 to R.X.], the National Natural Science Foundation of China [12371286, 11971039 to R.X.], the Postdoctoral Scientific Research Developmental Fund of Heilongjiang Province [LBH-Q18088 to S.M.], the fund of Beijing Information Science and Technology University [2025036 to Y.X.], the China Postdoctoral Science Foundation [2022M720308 to Z.J.], and Sino-Russian Mathematics Center. Part of the analysis was performed on the high-performance computing platform of the Center for Life Sciences (Peking University).

## Author contributions

R.X. and S.M. conceived the project. Y.X., R.X., and C.Z. designed the algorithm. Y.X. and J.L. designed the experiments. J.L., L.G., and B.J. performed the experiments. Y.X., Z.J., L.O., Y.D., and Y.S. analyzed the data. Y.X. and R.X. supervised the work. R.X., Y.X., Z.J., Y.D., L.O., and Y.S. wrote the manuscript.

## Competing interests

The authors declare the following competing interests: R.X. holds the stock of GeneX Health Co., Ltd. The remaining authors declare no competing interests.
