## [Peer Review File · Nature Communications]

TAGET: A toolkit for analyzing full-length transcripts from long-read sequencingREVIEWER COMMENTS

Reviewer #1 (Remarks to the Author):

Xia et al described TAGET, a pipeline for reconstructing transcript sequences from long-read RNA-seq data. It takes long reads as input, calls a long-read RNA-seq mapper, filters alignments, and reports high-quality transcripts. These post-alignment steps are necessary as the initial read alignment may be inaccurate. A key advance in TAGET is the use of CNN to identify accurate splice sites. TAGET could be a useful tool for processing long RNA-seq reads. Critical observations are as follows:

1) Evaluating sensitivity. The authors showed that TAGET achieves high specificity on transcript reconstruction. It is important to evaluate sensitivity at the same time. Achieving high specificity at a great cost of sensitivity would still be a problem. I understand that evaluating sensitivity on real data is tricky. Here are a couple of ideas. The authors may randomly remove X% of annotated isoforms and see how many of these isoforms could be reconstructed. Alternatively, they may consider simulation, like what StringTie2 paper was doing. Along this line, the authors should report the number of junctions or transcripts when talking about their accuracy (e.g. around line 137, 144, 189 and 193 among others). Such a number is not equivalent to sensitivity but it could be a rough proxy to sensitivity.

2) How would incomplete gene annotation affect the accuracy of TAGET? Human probably has more complete gene annotation than other model organisms. In my opinion, it is important to let readers know the effect of gene annotations. The authors may randomly remove X% of annotated isoforms and evaluate junction accuracy, number of novel genes, etc. This experiment would be similar to the one I suggested in 1).

3) Comparison to other tools on transcript reconstruction. TAGET seems to be inspired by SQANTI, so the authors only compared TAGET to SQANTI. I found it interesting that there are multiple other tools for transcript reconstruction, such as StringTie2, FLAIR, Espresso and IsoQuant, but they are not compared to SQANTI. It would be good to compare to a classical tool like StringTie2 unless the authors have a good argument against it. I predict StringTie2 will have higher sensitivity but a lot lower specificity.

4) Use minimap2 with known junctions. I could not find detailed command lines used for alignment. I guess the authors invoked minimap2 without the --junc-bed option. This option asks minimap2 to put more weight on annotated introns/junctions. This will improve the junction accuracy. I would recommend to use this option as TAGET relies on known annotations anyway.

5) About the three-way gene fusion. HLA-A has high diversity and has multiple pseudogenes including HLA-H. I wonder if a diverged allele of HLA-A may cause some mapping artifacts. It would be good if the authors could provide more evidence. Have the authors seen two HLA-A alleles in addition to this fusion on Transcript/23583? What is the genotype of HLA-A? Could the authors align all HLA-A alleles in IPD-IMGT/HLA to this transcript to confirm if it does not match any known HLA-A alleles?

Minor comment

6) The definition of FSM (Full Splice Match) was originated from SQANTI and has been often used by others. The authors are using a more stringent definition in TAGET. I can understand the motivation (especially on intron retention), but I would recommend the authors to follow the SQANTI definition to avoid confusion. The authors may add another category (e.g. FSM-stringent) for their current top tier.

Reviewer #2 (Remarks to the Author):

The transcript data from the third generation sequencing platforms can provide much longer reads than traditional RNA-seq data, and are becoming increasingly more popular. Therefore, it is quite necessary to develop appropriate computational tools for analyzing these full-length transcript data. Xia and his colleagues developed a toolkit called TAGET, which can perform a suite of computational analyses on full-length transcript data, ranging from transcript alignment to downstream expression analyses. The enclosed alignment algorithm is especially novel and appears to be significantly better than current available methods. The authors performed TAGET on both public and their own datasets, and further validated a number of their findings by experiments. This toolkit is timely and of potential high significance and I would like to see it published. Detailed comments and suggestions are listed below.

1. The authors showed that the CNN model could significantly improve the splice junction prediction. A comparison of two types of CNN models, one-dimensional model vs. two-dimensional one, should be performed.

2. TAGET cuts long reads to several overlapping short reads, aligns long reads and short reads with different mapping tools, and combine the two types of mapping results together. It seems that the choice of the short read length may influence the alignment quality. If you cut the long reads to short reads of different lengths, does the alignment accuracy change significantly?

3. Authors should compare the computational time of TAGET with alternative long read mapping tools. Is TAGET slower than the other tools because of it perform alignment for both long and short reads?

4. To which extend is the performance of expression quantification influenced by the sequencing coverage? How the consistency of the gene expression of the replicated samples changes with the sequencing coverage?

5. Figure 4 shows that the expressions given by Iso-seq are highly correlated between samples (>0.8 between LGC samples and > 0.98 between the OS replicates). Similarly, ONT expressions are also highly correlated (Figure 5). However, Figure S5 shows that the correlations between Iso-seq and RNA-seq expressions, though still positive, are in a lesser degree (~ 0.5). Please explain this phenomenon.

Reviewer #3 (Remarks to the Author):

The deep sequencing of transcripts may find some important isoforms during RNA splicing, which may control cell fate in either normal tissues or disease organs of human body, but most frequently the most abundant isoforms do commonly play dominant roles in cells as their translated proteins are overwhelming compared to the alternatively spliced isoform proteins.

This paper by Xiao et al. reported a toolkit TAGET as a very useful informatic tool for deep analysis of spliced isoforms from RNA-seq. They declare that:

1. TAGET is an integrative toolkit for transcript alignment, annotation, and expression quantification.
2. TAGET Enables Accurate Transcript Mapping in Iso-seq data.
3. TAGET Provides Accurate Transcript Annotation and Splice Junction Prediction.
4. TAGET Accurately Detects Gene Fusions with high sensitivity.
5. TAGET provides accurate gene expression quantification and identifies ECM1 as a DIU gene.
6. TAGET provides accurate alignment, annotation, and expression quantification for ONT data.

They last wrote as "our results strongly validated the performance of TAGET on Iso-seq data".

However, I did not see many data to validate the performance. As we know ECM1 contains 4 isoforms, but most people work on ECM1a because this isoform protein is secretory and highly oncogenic, while

ECM1b is non-secretory and plays a tumor suppression role at least in some specific cell lines and tissues reported by Yin et al. Nat Commun. 2021 Jul 9;12(1):4230.

All informatically calculated data or computer-based data must be carefully validated via both in vitro and in vivo experiments.

While this article does not match such standards to be accepted for publication, more data using TARGET to analyze the identified RNA isoforms from cell lines and tissues of at least 3 different cancers are needed. More detailed validation experiments must be designed to validate their computational isoform rightness.

Responses to Reviewers' Comments

We would like to thank the reviewers for their insightful comments and suggestions. We list our detailed point-to-point responses below. The comments are shown in italic and our responses are shown in blue. Changes in the manuscripts are shown in blue.

Reviewer 1:

Overall: Xia et al described TAGET, a pipeline for reconstructing transcript sequences from long-read RNA-seq data. It takes long reads as input, calls a long-read RNA-seq mapper, filters alignments, and reports high-quality transcripts. These post-alignment steps are necessary as the initial read alignment may be inaccurate. A key advance in TAGET is the use of CNN to identify accurate splice sites. TAGET could be a useful tool for processing long RNA-seq reads.

Response: We highly appreciate your careful review and helpful comments to help us improve the paper. Following your suggestions, we designed new experiments to evaluate the performance of TAGET and made changes in the TAGET pipeline. Please see below for details.

1) Evaluating sensitivity. The authors showed that TAGET achieves high specificity on transcript reconstruction. It is important to evaluate sensitivity at the same time. Achieving high specificity at a great cost of specificity would still be a problem. I understand that evaluating sensitivity on real data is tricky. Here are a couple of ideas. The authors may randomly remove X% of annotated isoforms and see how many of these isoforms could be reconstructed. Alternatively, they may consider simulation, like what StringTie2 paper was doing. Along this line, the authors should report the number of junctions or transcripts when talking about their accuracy (e.g., around line 137, 144, 189 and 193 among others). Such a number is not equivalent to sensitivity but it could be a rough proxy to sensitivity.

Response: Thanks for your constructive comment. We evaluated the sensitivity of TAGET and compared with SQANTI as well as the three algorithms in the reviewer's comment 3 including IsoQuant, StringTie2, and FLAIR. Following the reviewer's suggestion, we randomly deleted annotations of X% (X = 5, 10, 20, 30, and 50) of genes and evaluated how many deleted junctions could be recovered. More specifically, for a given dataset, we first ran these algorithms using the complete annotation as the reference database, and took the junctions of the deleted genes that were reported by any of the five methods as the set of positives. We then ran each tool using the incomplete annotation (i.e., the annotation without the deleted genes) and calculated the number of correctly recalled junctions in the set of positives (true positives). We found that TAGET and SQANTI achieved higher sensitivities (the ratio between the numbers of true positives and positives) than IsoQuant, StringTie2 and FLAIR (Figure R1A). Due to their extensive filtering, StringTie2 and FLAIR were very conservative and had much lower sensitivities compared with the other tools, which was consistent with the observation that they detected much fewer novel junctions (Figure R1B). The precisions of StringTie2 and FLAIR, as measured by the proportion of canonical junctions in the novel junctions, were higher than other methods (Figure R1C. Please also see our responses to comment 2 and 3). Overall, TAGET achieved a very high sensitivity but also a reasonable high precision.

Following the reviewer's suggestion, we also reported the numbers of junctions given by TAGET, SQANTI, IsoQuant, StringTie2, and FLAIR (Figure R1D and Supplementary Fig 4). Consistently, we found that TAGET, SQANTI, and IsoQuant generally reported more junctions than StringTie2 and FLAIR. We also presented the numbers of junctions reported by the alignment tools TAGET, TAGET-w/o, minimap2, and GMAP in Supplementary Fig 3B. Supplementary Table 2-8 provide more detailed information about the number of transcripts and junctions. Finally, note that TAGET and SQANTI are transcript alignment tools, while IsoQuant, StringTie2, and FLAIR are tools for transcript reconstruction based on assembly. In addition to

alignment, TAGET could also perform downstream analyses such as gene fusion detection, transcript expression quantification, and differentiable isoform usage analyses.

Figure R1: **(A)** Mean sensitivities of different algorithms over 11 samples using annotations with different percentages of genes deleted. **(B)** The number of novel junctions reported using the complete annotation (the numbers are shown in the log10 scale). **(C)** The proportions of canonical junctions in novel junctions. **(D)** The numbers of unique junctions reported by different algorithms for the different datasets.

2) How would incomplete gene annotation affect the accuracy of TAGET? Human probably has more complete gene annotation than other model organisms. In my opinion, it is important to let readers know the effect of gene annotations. The authors may randomly remove X% of annotated isoforms and evaluate junction accuracy, number of novel genes, etc. This experiment would be similar to the one I suggested in 1).

Response: Following the reviewer’s suggestion, we randomly removed 5%, 10%, 20%, 30%, and 50% of genes from the annotation file to evaluate the performance of different algorithms. Sensitivities were shown in Figure R1A. We further investigated the influence of the completeness of gene annotation on junction precision and novel gene detection. The junction precision is defined as the proportion of true junctions in all discovered junctions. True junctions are the junctions in the annotation database. Overall, the completeness of gene annotation did not have much influence on the precisions of the different algorithms (Figure R2A). Consistent with Figure R1C, StringTie2 and FLAIR had higher precisions than other algorithms, at the cost of the significant reduction of their sensitivities. As suggested by the reviewer, we further investigated the number of true novel genes detected by different methods (Figure R2B). True novel genes are the detected genes in the deleted annotations. The number of true novel genes is a rough proxy to sensitivity. Again, we see that TAGET and SQANTI achieved higher sensitivities than IsoQuant, StringTie2 and FLAIR. Further, we found that the completeness of gene annotation did not have significant impact on the proportions of canonical junctions in novel junctions (Figure R2C).

Figure R2: **(A)** Mean precisions of different algorithms over 11 samples using different incomplete annotations. **(B)** The number of novel genes reported using different incomplete annotations. **(C)** The proportions of canonical junctions in novel junctions over 11 samples using different incomplete annotations.

3) Comparison to other tools on transcript reconstruction. TAGET seems to be inspired by SQANTI, so the authors only compared TAGET to SQANTI. I found it interesting that there are multiple other tools for transcript reconstruction, such as StringTie2, FLAIR, Espresso and IsoQuant, but they are not compared to SQANTI. It would be good to compare to a classical tool like StringTie2 unless the authors have a good argument against it. I predict StringTie2 will have higher sensitivity but a lot lower specificity.

Response: Following the reviewer's suggestion, we compared with IsoQuant, FLAIR, and StringTie2 (see our responses to the reviewer's comment 1 and 2). Overall, TAGET achieved high sensitivities and precisions. The precisions of StringTie2 and FLAIR were higher than TAGET, but had considerable lower sensitivities. The reason was that StringTie2 and FLAIR reconstruct longer transcripts from multiple reads and had stringent filtering to remove potential false positives, leading to their high precisions but low sensitivities. In contrast, TAGET is a toolkit for transcript alignment, annotation, and quantification that seeks the best alignment for each transcript without stringent read filtering, making TAGET much more sensitive than StringTie2 and FLAIR while still having reasonable high precisions. We also added more discussion about the transcript reconstruction tools StringTie2, FLAIR, Espresso, and IsoQuant in the manuscript (page 14).

4) Use minimap2 with known junctions. I could not find detailed command lines used for alignment. I guess the authors invoked minimap2 without the --junc-bed option. This option asks minimap2 to put more weight on annotated introns/junctions. This will improve the junction accuracy. I would recommend to use this option as TAGET relies on known annotations anyway.

Response: Thanks for the suggestion. By default, we now use the "--junc-bed" option for minimap2 alignment. All analyses in the paper are now performed with the option. The junctions reported by minimap2 with and without this parameter only have subtle differences (Figure R3) and all conclusions in the paper remained the same. We added the "--junc-bed" parameter in config file of TAGET so that TAGET can run minimap2 with this parameter.

Figure R3: The distribution of splice junction biases using minimap2 with and without the “--junc-bed” option. The corresponding percentages of predicted splice junctions in different bias categories are displayed.

5) About the three-way gene fusion. HLA-A has high diversity and has multiple pseudogenes including HLA-H. I wonder if a diverged allele of HLA-A may cause some mapping artifacts. It would be good if the authors could provide more evidence. Have the authors seen two HLA-A alleles in addition to this fusion on Transcript/23583? What is the genotype of HLA-A? Could the authors align all HLA-A alleles in IPD-IMGT/HLA to this transcript to confirm if it does not match any known HLA-A alleles?

Response: Thanks for raising this important problem. Following the reviewer’s suggestion, we determined the HLA-A genotype of this patient using the seq2HLA method. The patient is heterozygous in HLA-A with two alleles HLA-A03:01 and HLA-A24:02. We aligned transcript/23853 to the sequences of HLA-A03:01 and HLA-A24:02 in the IPD-IMGT/HLA database. We found that transcript/23853 could be aligned to HLA-A24:02:156 with one alignment gap suggesting that transcript/23853 was likely derived from HLA-A24:02 itself rather than from a three-way gene fusion event. This false positive was generated because the annotation database used by TAGET contained very few HLA allele sequences. We removed this three-way gene fusion in the manuscript. To avoid similar false positives, we add a filter to filter HLA related fusions in the TAGET pipeline.

Minor comment:

6) The definition of FSM (Full Splice Match) was originated from SQANTI and has been often used by others. The authors are using a more stringent definition in TAGET. I can understand the motivation (especially on intron retention), but I would recommend the authors to follow the SQANTI definition to avoid confusion. The authors may add another category (e.g., FSM-stringent) for their current top tier.

Response: Thank you for your suggestion. We followed the definition of FSM from SQANTI and calculated the proportion of each category (Figure R4). We found that the FSMs reported by TAGET were now very

similar to SQANTI’s result. In the output of TAGET, we now use the SQANTI’s definition of FSM and mark those satisfying TAGET’s definition as FSM-stringent. However, in the paper, since we believe that TAGET’s definition of FSM is more appropriate, all figures in the paper still use TAGET’s definition.

Figure R4: The distributions of the seven transcript types annotated by TAGET (bars on the left) and SQANTI (bars on the right). **(A)** TAGET uses SQANTI’s definition of FSM. **(B)** TAGET uses TAGET’s definition of FSM.

Reviewer #2 (Remarks to the Author):

Overall: The transcript data from the third-generation sequencing platforms can provide much longer reads than traditional RNA-seq data, and are becoming increasingly more popular. Therefore, it is quite necessary to develop appropriate computational tools for analyzing these full-length transcript data. Xia and his colleagues developed a toolkit called TAGET, which can perform a suite of computational analyses on full-length transcript data, ranging from transcript alignment to downstream expression analyses. The enclosed alignment algorithm is especially novel and appears to be significantly better than current available methods. The authors performed TAGET on both public and their own datasets, and further validated a number of their findings by experiments. This toolkit is timely and of potential high significance and I would like to see it published. Detailed comments and suggestions are listed below.

Response: We sincerely thank you for your review and encouraging comments. We made detailed explanations and designed the experiments to address the comments below.

1. The authors showed that the CNN model could significantly improve the splice junction prediction. A comparison of two types of CNN models, one-dimensional model vs. two-dimensional one, should be performed.

Response: Following the reviewer’s comment, we also compared with a one-dimensional CNN model. The CNN model consists of three layers: a convolutional layer, a pooling layer by the rectified linear units (ReLU) and a batch-normalization layer. The inputs of the CNN model are 41 bp sequences which consists of 21bp sequences near splice sites (both exon and intron sequences) and its 10 bp upstream and 10 bp downstream sequences. The outputs are the splice positions of the sequences. Given a input sequence S of length $n(n=41)$, We encoded it to a padded 5-row array that each row represents one of four bases (A, G, T, C) or the splice site. Specifically, given the size of convolution kernel $m(m=11)$, we convert S to a $5 * (n + 2m - 2)$ array A in this way

$$A_{i,j} = \begin{cases} 0.25, & i = 1,2,3,4 \text{ and } j < m \text{ or } j > n - m \\ 1, & (i, S_{j-m+1}) = (1, A) \text{ or } (2, T) \text{ or } (3, G) \text{ or } (4, C) \\ 1, & i = 5 \text{ and the splice site is } j - m + 1 \\ 0, & \text{otherwise} \end{cases},$$

where $A_{i,j}$ is the (i, j) element of A and S_i denotes the i -th base of S .

We built a one-dimensional CNN model to perform the local adjustment. The convolution kernels are chosen as 2048 kernels of length 11 with 5-channels. The loss function is chosen as the categorical cross-entropy loss and the Adam optimizer is used to minimize the loss function. The model is trained for 1000 epochs on NVIDIA GeForce GTX 1080 Ti GPUs. The learning rate of the optimizer is set to 0.01. This procedure is repeated 10 times to make the model more accurate. Figure R5 shows the distribution of splice site biases of TAGET with the original CNN model and the one-dimensional CNN model (TAGET-1) as well as other alignment tools. We see that TAGET and TAGET-1 performed very similarly (Table R1), and TAGET gave slightly better alignment than TAGET-1.

Figure R5: The distribution of splice site biases of TAGET, TAGET-1 (TAGET with the one-dimensional CNN model), TAGET-w/o (TAGET without CNN), GMAP, and minimap2.

Table R1: The proportions of splice sites with no bias given by TAGET and TAGET-1 (TAGET with one-dimensional CNN model). TAGET performed slightly better than TAGET-1 on the sample LGC-265C and LGC-415N.

Dataset	LGC-133C	LGC-133N	LGC-265C	LGC-265N	LGC-415C	LGC-415N	OS-1-C	OS-1-N	OS-2-C	OS-2-N
TAGET	0.9838995	0.9826437	0.9745733	0.9605888	0.9605536	0.9507573	0.9730890	0.9774652	0.9703983	0.9774508
TAGET-1	0.9838995	0.9826437	0.9745302	0.9605888	0.9605536	0.9507500	0.9730890	0.9774652	0.9703983	0.9774508

2. TAGET cuts long reads to several overlapping short reads, aligns long reads and short reads with different mapping tools, and combines the two types of mapping results together. It seems that the choice of the short read length may influence the alignment quality. If you cut the long reads to short reads of different lengths, does the alignment accuracy change significantly?

Response: We varied the short read length and slide window size and evaluated the alignment accuracy and computational time of TAGET with different combinations of the short read length and slide window length (Figure R6). The alignment accuracy is defined as the proportion of splice junctions matching the reference. Overall, as expected, when the slide window became larger, TAGET was computationally more efficient. In terms of alignment accuracy, longer short reads tended to give better alignment. Overall, the default parameter value of TAGET with the short read length 250 bp and slide window size 100bp gave high alignment accuracy and also had good computational efficiency.

Figure R6: The alignment accuracy (the green bubbles) and the running time (the bars) of TAGET when using different combinations of short read length and slide window size. The numbers below the bars are read lengths and slide window sizes. For example, 50_10 means that read length is 50 bp and slide window size is 10 bp.

3. Authors should compare the computational time of TAGET with alternative long read mapping tools. Is TAGET slower than the other tools because of it perform alignment for both long and short reads?

Response: Table R2 shows the computational time of TAGET, minimap2 and GMAP. TAGET is computationally more demanding than minimap2 and GMAP, because TAGET alignment is based on long read and short read mapping alignment. However, TAGET was still relatively fast and could finish alignment in 30 minutes with 8 CPUs for the samples in our manuscript.

Table R2: The computational time (in seconds) of TAGET and SQANTI2.

Dataset	LGC-133C	LGC-133N	LGC-265C	LGC-265N	LGC-415C	LGC-415N	OS-1-C	OS-1-N	OS-2-C	OS-2-N
TAGET	1375	1203	1622	1577	1601	1723	1184	1876	1955	2609
minimap2	602	563	750	703	742	771	659	869	803	1120
GMAP	1293	1125	1602	1536	1550	1842	1009	1722	1885	2350

4. To which extend is the performance of expression quantification influenced by the sequencing coverage? How the consistency of the gene expression of the replicated samples changes with the sequencing coverage?

Response: We down-sampled the data to study the influence of sequencing coverage on the accuracy of expression quantification (Figure R7). The accuracy is evaluated as the correlation between expression given by down-sampled data and original data. We found that when the library size was over 3G, the performance of expression quantification was largely plateaued; while when library size was less than 3G, expression quantification started to deteriorate.

Figure R7: The correlation of expression between two samples with different library sizes.

5. Figure 4 shows that the expressions given by Iso-seq are highly correlated between samples (>0.8 between LGC samples and > 0.98 between the OS replicates). Similarly, ONT expressions are also highly correlated (Figure 5). However, Figure S5 shows that the correlations between Iso-seq and RNA-seq expressions, though still positive, are in a lesser degree (~ 0.5). Please explain this phenomenon.

Response: The high correlations between Iso-seq samples and replicates demonstrated the high reproducibility of TAGET's expression quantification. The weaker correlations between Iso-seq and RNA-seq might be due to following reasons. (1) RNA-seq or Iso-seq might have platform-specific biases. For example, RNA-seq often involves a PCR amplification step. It is well-known that the efficiency of PCR amplification depends on the sequence contexts such as the GC-content. Thus, the RNA-seq expression might have biases due to PCR amplification. On the other hand, Iso-seq is single molecular sequencing and does not involve PCR amplification and thus does not have amplification biases. (2) Compared with long-reads, short reads from one gene are more likely to be misassigned to other genes because of the sequence similarity between genes, and the incorrect assignment of short reads to genes would lead to biased expression quantification.

Reviewer #3 (Remarks to the Author):

Overall: The deep sequencing of transcripts may find some important isoforms during RNA splicing, which may control cell fate in either normal tissues or disease organs of human body, but most frequently the most abundant isoforms do commonly play dominant roles in cells as their translated proteins are overwhelming compared to the alternatively spliced isoform proteins.

This paper by Xiao et al. reported a toolkit TAGET as a very useful informatic tool for deep analysis of spliced isoforms from RNA-seq. They declare that:

1. TAGET is an integrative toolkit for transcript alignment, annotation, and expression quantification.
2. TAGET Enables Accurate Transcript Mapping in Iso-seq data.
3. TAGET Provides Accurate Transcript Annotation and Splice Junction Prediction.
4. TAGET Accurately Detects Gene Fusions with high sensitivity.
5. TAGET provides accurate gene expression quantification and identifies ECM1 as a DIU gene.
6. TAGET provides accurate alignment, annotation, and expression quantification for ONT data.

Response: We appreciate the time and effort you put into reviewing our manuscript and providing us with valuable feedback. Below, we address these comments point by point.

They last wrote as” our results strongly validated the performance of TAGET on Iso-seq data”. However, I did not see many data to validate the performance. As we know ECM1 contains 4 isoforms, but most people work on ECM1a because this isoform protein is secretory and highly oncogenic, while ECM1b is non-secretory and plays a tumor suppression role at least in some specific cell lines and tissues reported by Yin *et al. Nat Commun. 2021 Jul 9;12(1):4230.*

Response: Thank you for bringing this work (Yin *et al. Nat Commun. 2021*) to our attention. We used three additional cell lines to further evaluate the effects of ECM1a (ECM1-202) and ECM1b (ECM1-201) on cell migration and invasion (Figure R8). The three cell lines were HeyA8 (a human ovarian cancer cell line used by Yin *et al.*), AU565 (a human breast cancer cell line) and 5-8F (another nasopharyngeal carcinoma cell line).

Figure R8: Effects of ECM1-201 and ECM1-202 isoforms on cell migration and invasion using the Transwell system. Three cell lines were tested: HeyA8 (A), AU565 (B) and 5-8F (C) (Error bars: mean \pm SD. **: Student's t-test, $p < 0.01$; ***: Student's t-test, $p < 0.001$).

Consistent with the findings of Yin et al. Nat Commun. 2021, in the HeyA8 and AU565 cell lines, ECM1b significantly inhibited cell migration and invasion, while ECM1a significantly promoted cell migration and invasion, indicating the tumor suppression role of ECM1b and oncogenic role of ECM1a. In the 5-8F cell line, ECM1b still showed significant inhibition effect on cell migration and invasion, but ECM1a did not show any significant effect, which was consistent with the results of the nasopharyngeal carcinoma cell line 6-10B considered in the first submission of this paper. These results suggested different roles of the two ECM1 isoforms in various cancer types, consistent with our finding that ECM1 was a DIU gene between tumor and normal tissues in the three LGC samples.

Furthermore, we note that this manuscript is to present the computational tool TAGET for analyzing full-length transcriptome data. The validations of TAGET included both experimental and computational validations. Many merits of TAGET were comprehensively validated by computational validations. For example, the splice junction predictions were computationally validated by analyzing sequence features of junction sites and by comparing with corresponding RNA-seq data; gene fusion detection and gene expression quantification were computationally validated using RNA-seq data; Simulations were also performed to evaluate the performance of TAGET on junction prediction and gene fusion detection.

All informatically calculated data or computer-based data must be carefully validated via both in vitro and in vivo experiments.

While this article does not match such standards to be accepted for publication, more data using TAGET to analyze the identified RNA isoforms from cell lines and tissues of at least 3 different cancers are needed. More detailed validation experiments must be designed to validate their computational isoform rightness.

Response: Following the reviewer's suggestion, we further sequenced Iso-seq data from three pairs of lung squamous cell carcinoma samples. Now, we had Iso-seq data from 4 different cancer types including lung squamous cell carcinoma (3 pairs of tumor and normal), laryngocarcinoma (3 pairs), osteosarcoma (1 pair with 2 replicates) and a public melanoma cell line (COLO829). We incorporated the newly sequenced Iso-seq data and updated all analyses in the paper. We further randomly selected 33 novel transcripts and 12 gene fusions detected from the lung cancer samples for validation, and found that 29 novel transcripts (87.9%) and 11 gene fusions (91.7%) were validated by Sanger sequencing. In total, we had 73 novel transcripts (33 from the new samples and 40 from the original samples) and 35 fusions (12 from the new samples and 23 from the original samples) for validation. The overall validation rates were 86.5% (63/73) and 85.7% (30/35) for the novel transcripts and gene fusions, respectively.

We further validated DIU genes by qRT-PCR (Figure R9). We selected 20 DIU genes from the LGC samples for validation and were only successfully designed primers for two of the candidate DIU genes. Consistent with TAGET's prediction (Figure R10), we found that the expression of the isoform TPM4-201 was significantly increased in cancer samples, whereas the isoform TPM4-202 showed comparable expression levels in both cancer and normal samples. The isoform PLS3-202 showed increased expression in the tumor sample of LGC-133, whereas the isoform PLS3-209 had decreased expression. In total, we validated 7 gene-sample pairs (TPM4 in 3 samples, PLS3 in 1 sample and ECM1 in 3 samples) and the DIU events were validated.

Figure R9: Comparison of expression levels from qRT-PCR between TPM4-201 and TPM4-202 (A), as well as PLS3-202 and PLS3-209 (B) in LGC cancer and normal samples (Error bars: mean \pm SD).

Figure R10: Comparison of expression levels from TAGET between TPM4-201 and TPM4-202 (A), as well as PLS3-202 and PLS3-209 (B) in LGC cancer and normal samples.

REVIEWERS' COMMENTS

Reviewer #1 (Remarks to the Author):

The authors have addressed most of my concerns. It is good to see TAGET more accurate than IsoQuant and StringTie2. I still have a couple of comments.

1) I am a little confused by the authors' response and would appreciate a clarification here. The authors said "TAGET and SQANTI are transcript alignment tools, while IsoQuant, StringTie2, and FLAIR are tools for transcript reconstruction based on assembly". In my understanding, SQANTI and TAGET take existing long-read alignments, improve them and report a set of non-redundant transcripts. Is it correct? If so, TAGET and SQANTI would be closer to IsoQuant and StringTie2 that aim to reconstruct a transcriptome based on read alignment. These are not "alignment tools".

2) Along this line, when comparing TAGET with minimap2/GMAP (e.g. around Line 142 and in Fig S3B), how did the authors count "predicted junctions"? Minimap2/GMAP aligns individual reads. One junction is often supported by multiple reads. Did the authors count this once or multiple times? Meanwhile, the README of TAGET doesn't mention BAM as an output. Does TAGET only report one "predicted junction" even if it is supported by multiple reads?

Reviewer #2 (Remarks to the Author):

All my concerns have been addressed and the manuscript is suitable for publication.

Reviewer #3 (Remarks to the Author):

The authors performed more experiments and addressed my comments exactly. I am satisfied with the response.

Responses to Reviewers' Comments

We are very grateful for your review. And we list the point-to-point response below. The comments are shown in italic and our responses are shown in blue.

1) I am a little confused by the authors' response and would appreciate a clarification here. The authors said "TAGET and SQANTI are transcript alignment tools, while IsoQuant, StringTie2, and FLAIR are tools for transcript reconstruction based on assembly". In my understanding, SQANTI and TAGET take existing long-read alignments, improve them and report a set of non-redundant transcripts. Is it correct? If so, TAGET and SQANTI would be closer to IsoQuant and StringTie2 that aim to reconstruct a transcriptome based on read alignment. These are not "alignment tools".

Response: Sorry about the confusion. The description in the previous response was not accurate. TAGET has a step to integrate alignment results from short read alignment tools and long read alignment tools for more accurate alignments and thus TAGET could be viewed as an alignment tool, while SQANTI directly takes long-read alignments for downstream analyses and is not an alignment tool. After alignment, TAGET further performs annotation, transcript quantification and gene fusion detection analyses. Finally, TAGET obtains non-redundant transcripts and their expressions. In comparison, IsoQuant and StringTie2 utilize multiple reads from the same gene to generate a non-redundant set of transcripts. In terms of transcript identification, TAGET and SQANTI are similar to IsoQuant and StringTie2, but TAGET can also perform alignment as well as transcript quantification and gene fusion analyses.

2) Along this line, when comparing TAGET with minimap2/GMAP (e.g. around Line 142 and in Fig S3B), how did the authors count "predicted junctions"? Minimap2/GMAP aligns individual reads. One junction is often supported by multiple reads. Did the authors count this once or multiple times? Meanwhile, the README of TAGET doesn't mention BAM as an output. Does TAGET only report one "predicted junction" even if it is supported by multiple reads?

Response: When we compare the alignments (e.g. Fig 2B and Fig S3B), because the focus is on the alignment quality of the long-reads, we investigate the junctions predicted from each long-reads and thus junctions supported by multiple reads can be counted multiple times. Figure 2B shows the accuracy of different software alignments, while Figure S3B displays the number of reported junctions by different software when comparing accuracy. TAGET does not output a BAM file. The alignment output of TAGET is read-centric, that is, TAGET outputs each read's alignment and its annotations separately (in a bed file). Consequently, junctions supported by multiple reads will be reported multiple times. At the same time, TAGET also outputs the expression files of each gene and transcript, and a gene fusion file.